# Handling Associated with Drenching Does Not Impact Survival and General Health of Low Birth Weight Piglets

**DOI:** 10.3390/ani11020404

**Published:** 2021-02-05

**Authors:** Kevin Van Tichelen, Sara Prims, Miriam Ayuso, Céline Van Kerschaver, Mario Vandaele, Jeroen Degroote, Steven Van Cruchten, Joris Michiels, Chris Van Ginneken

**Affiliations:** 1Applied Veterinary Morphology, Faculty of Biomedical, Pharmaceutical and Veterinary Sciences, Antwerp University, Universiteitsplein 1, 2610 Wilrijk, Belgium; kevin.vantichelen@uantwerpen.be (K.V.T.); sara.prims@uantwerpen.be (S.P.); miriam.ayusohernando@uantwerpen.be (M.A.); steven.vancruchten@uantwerpen.be (S.V.C.); 2Laboratory for Animal Production and Animal Product Quality, Faculty of Bioscience Engineering, Ghent University, Coupure Links 653, 9000 Ghent, Belgium; celine.vankerschaver@ugent.be (C.V.K.); mario.vandaele@ugent.be (M.V.); jerdgroo.degroote@ugent.be (J.D.); joris.michiels@ugent.be (J.M.)

**Keywords:** pig, performance, management, oral supplementation, neonatal, mortality

## Abstract

**Simple Summary:**

In the modern pig industry, one of the main goals has been to increase litter sizes, and thus, improve sows’ production efficiency. However, this increase in litter size has also resulted in an elevated number of low birth weight piglets and a higher perinatal mortality. In an attempt to reduce mortality in low birth weight piglets, many studies focus on drenching bioactive substances. However, most studies only focus on the supplement and neglect any potential effect of drenching itself. Given that low birth weight piglets are often very weak, drenching might provoke additional stress, and consequently, nullify the effect of the supplement or even negatively affect the piglet’s health. In this study, low birth weight piglets were sham drenched by conducting the drenching act without a liquid (putting an empty syringe into the animal’s mouth) to evaluate the effect of drenching on their body weight, health and mortality. No negative or positive effect of drenching was observed, and thus, it was concluded that drenching is a safe tool that can be implemented in good pre-weaning management. However, studies that examine the oral supplementation of bioactive substances should always include a sham and a negative control group to ensure that the observed results can be attributed to the supplement, rather than the act of drenching.

**Abstract:**

The increase in litter sizes in recent years has resulted in more low birth weight (LBW) piglets, accompanied by a higher mortality. A potential intervention to overcome this is drenching bioactive substances. However, if the act of drenching provokes additional stress in LBW piglets, it might counteract the supplement’s effect and be detrimental for the piglet’s survival. To study the effect of the drenching act, piglets from 67 sows were weighed within 4 h after birth. The mean litter birth weight (MLBW) and standard deviation (SD) were calculated. LBW piglets (*n* = 76) were defined as weighing between (MLBW-1*SD) and (MLBW-2.5*SD). They were randomly allocated to two treatments: “sham” (conducting the act of drenching by inserting an empty 2.5 mL syringe in the mouth during 20 s, once a day, d1 till d7; *n* = 37) or “no treatment” (no handling; *n* = 39). On day 1, 3, 9, 24 and 38, piglets were weighed and scored for skin lesions. Blood samples were collected on day 9 and 38 and analyzed to determine glucose, non-esterified fatty acids (NEFA), urea, immunoglobulin G (IgG), insulin-like growth factor 1 (IGF-1) and a standard blood panel test. There was no difference between sham drenched and untreated piglets regarding any of the parameters. In conclusion, this study showed that drenching does not impose a significant risk to LBW piglets and can be applied safely during the first 7 days after birth.

## 1. Introduction

During the last three decades, the pig industry has shown an increased interest in larger litter sizes. Through genetic selection and improved management, average litter size [1] and, consequently, the efficiency of pork production have increased [2,3]. Although this increase in litter size often leads to financial gains, it is associated with a higher morbidity and mortality during the perinatal period [4,5,6,7]. The least desirable outcome of an increased litter size is a larger proportion of piglets with a low birth weight (LBW) [8,9]. These LBW piglets are more susceptible to crushing [10,11], hypothermia [12,13], starvation [13,14], and thus, attribute greatly to increased pre-weaning mortality [15]. Over the past few years, different strategies have been implemented to improve the survival chances of all, but especially of LBW piglets, such as sow-fixating farrowing crates [16], split suckling [17,18] or cross-fostering [19]. Another potential intervention that might benefit LBW piglets’ survival is supplementary feeding to provide extra energy [20,21] or bioactive substances (colostrum, oligosaccharides, antioxidants, etc.) [22,23,24,25,26]. Piglets can be supplemented with (enriched) milk replacer or colostrum by providing a plastic dish, a bowl or a milk bar in the farrowing box [22,27]. More sophisticated devices, such as automated feeders, are available as well [28]. Although these systems can be a valuable asset to pre-weaning management, their use is often limited by a high labor cost (maintenance, refilling, cleaning) and high spillage of the used products [27]. What is more, these applications cannot be used to provide an individual or dose-dependent supplementation and are consequently also used by piglets that need no support. This might result in an insufficient nutritional intake by smaller piglets as a result of their competitive disadvantage [10,29]. A frequently used strategy to supplement individual piglets with a given amount of milk or substance in a certain dosage is drenching. However, most studies only focus on the effect of the supplemented product, whereas the effect of drenching itself is often neglected [20,21,23,30]. Drenching usually involves catching and fixating the animals, thus potentially acting as a stressor. Given the LBW piglets’ vulnerable condition, this potential stressor might further aggravate their chance of survival and negatively influence their health and survival, rather than provide the animals with additional energy or bioactive compounds. Consequently, it could be expected that sham drenching LBW piglets daily during their first week of life might result in higher mortality rates when these weakened animals are confronted with normal husbandry procedures that act as acute stressors, such as castration or teeth clipping. Therefore, it was hypothesized in this study that the act of drenching would provoke additional stress and affect LBW piglets negatively, resulting in a worsened performance (body weight, general (blood) health parameters) and a higher pre-weaning mortality.

## 2. Materials and Methods

### 2.1. Ethical Approval

This study was reviewed and approved by the Ethical Committee for Animal Experimentation of the University of Antwerp (ECD 11/2018) and was compliant with national legislation and European guidelines (2010/63/EC).

### 2.2. Animals

The experiments were conducted on a commercial farm in Meer (Hoogstraten, Belgium). All sows (Topigs20 (*n* = 58), Norwegian Landrace (*n* = 9)) were kept in individual farrowing crates of 2.25 × 0.60 m that were located in pens of 2.50 × 1.75 m. The parity of the sows varied from 1 to 10, with a mean parity of 4.35 ± 2.11 standard deviation (SD). The sows were fed with a commercial gestation diet up to farrowing. After farrowing, all sows were switched to a commercial lactation diet. Declared nutrient and chemical composition can be found in Table 1. Piglets included in the study, as well as their littermates, were subjected to the standard handling procedures in the farm: before the age of one week, all piglets were ear tagged, tail docked, received a 200 mg iron dextran injection and all male piglets were castrated using meloxicam analgesics. Piglets were weaned at the age of 3 weeks.

### 2.3. Piglet Selection

All piglets were weighed within 4 h after parturition. For each litter, the mean birth weight (MLBW) and SD were calculated. Piglets with a birth weight of MLBW-2.5*SD < piglet birth weight < MLBW-1*SD were characterized as “LBW piglets”. In each litter, a maximum of two LBW piglets was selected and ear tagged. In total, 76 LBW piglets were selected, spread over 6 farrowing rounds and 67 sows. The body weight, skin lesion score and mortality of the piglets were recorded on the day of birth (d1), on day 3 (d3), day 9 (d9), 2 days after weaning (d24) and 2 weeks after weaning (d38).

### 2.4. Experimental Treatments

To minimize the influence of sow effects and due to the large number of LBW piglets needed to observe a potential effect of drenching, treatments were allocated on piglet level rather than litter level. The LBW piglets were randomly allocated to a treatment: “sham” or “no treatment (none)”. The “sham” intervention implied a fake drenching by inserting an empty 2.5 mL syringe into the piglet’s mouth. This was repeated once a day during the first week after birth (d1 till d7). Based on preliminary testing with volumes of 2 mL of milk, the average catching and drenching time was 29.6 sec ± 8.1 SD per piglet (average catching time by 1 person: 10.5 sec ± 5.9; average drenching time: 19.0 sec ± 5.7). Therefore, every piglet was sham drenched during 20 s. Animals that belonged to the “no treatment” group were left in the pen and were not picked up nor drenched.

### 2.5. Data Collection

#### 2.5.1. Skin Lesion Scoring

Skin lesion scoring has been validated as an indicator for aggressive behavior [31]. It is often applied in studies that focus on supplementation or diets [32,33], following the hypothesis that the supplement might provide the piglet with additional energy, allowing it to demonstrate more competitive, aggressive behavior. To determine whether handling (drenching) LBW piglets had an effect on their competitive behavior, and consequently, to ensure that any observed difference in skin lesions during future supplementation studies can be attributed to the supplement and not the act of drenching, each LBW piglet was scored for skin lesions on the snout and skin.

A skin lesion score was given using the following scoring system according to Rundgren and Löfquist [34], Pluske and Williams [35] and Parrat et al. [36]:

0: no lesions

1: <5 superficial lesions (skin unbroken)

2: 5–10 superficial lesions or <5 deep lesions (skin broken and evidence of hemorrhage)

3: >10 superficial lesions or >5 deep lesions

The skin lesion scoring was performed on d1, d3, d9, d24 and d38.

#### 2.5.2. Blood Sampling

At the end of the drenching period (d9) and 2 weeks after weaning (d38), an 8 mL blood sample was taken from the cranial vena cava. For ethical reasons, no more than 2 attempts to draw blood were allowed. The blood sample was divided into 3 tubes: 1 serum tube, 1 ethylene diamine tetra acetic acid (EDTA) tube and 1 heparin tube. The serum and EDTA tubes were sent to Animal Health Care (Torhout, Belgium) for routine biochemical and hematological analysis. The following biochemical parameters were determined: glucose, non-esterified fatty acids (NEFA) and urea. The hematological analysis determined the levels of red blood cells (RBCs), the hematocrit (HCT), the hemoglobin (HGB) levels, the lymphocytes, monocytes, neutrophils, eosinophils, basophils, the total white blood cell (WBC) count and the platelet (thrombocyte) levels.

The heparin tube was centrifuged at 1500 g (3500 rpm) for 10 min at 4 °C. Next, the supernatant or plasma was collected and kept at −80 °C until further analysis.

#### 2.5.3. IgG and IGF-1 Analysis

The immunoglobulin G (IgG) and insulin-like growth factor 1 (IGF-1) levels were measured using a porcine competitive inhibition and a sandwich enzyme immunoassay, respectively (IgG: Cloud-Clone Corp., Katy, TX, USA, CEA544Po; IGF-1: Cloud-Clone Corp., Katy, TX, USA, SEA050Po). The collected plasma was diluted (1/2500 and 1/50, respectively) and IgG and IGF-1 levels were determined according to the manufacturer’s instructions. All samples were loaded in triplicate.

### 2.6. Statistical Analysis

To evaluate the potential effect of drenching on all outcome variables, linear mixed models were fitted in JMP Pro 14 (SAS Institute Inc., Cary, NC, USA). Treatment and age were added as fixed effects and sex was considered a covariate (with the exception of the IgG and IGF-1 analysis, due to a limited number of male animals (*n* = 3)). In addition, all 2-way interactions between treatment, age and sex were included. Interactions in third degree were not added, as these would have made the model too complex. Given the fact that the piglets were selected over a period of 10 months (6 selection rounds), the farrowing round was added as a random effect. To account for the dependence between littermates and the multiple measurements that were performed on the same piglets, the sow (nested in the farrowing round) and the piglet (nested in sow which was nested in the farrowing round) were included, respectively, as random effects as well. Sows which had been used for piglet selection during previous farrowing rounds were neglected, thus each sow was only included once. This starting model was simplified using stepwise backwards modelling, during which all non-significant effects were removed from the starting model. To meet normality and/or homoscedasticity, body weight, NEFA, urea, IgG, IGF-1 and neutrophil levels were log transformed, while the other outcome variables required no transformations. Effects were considered statistically significant if *p ≤* 0.05. Post-hoc analysis with Tukey’s correction was used to compare different groups. All values are presented as median ± SD. To evaluate the probability of more severe skin lesions occurring in certain treatment or age groups, an ordinal logistic regression model was used in which treatment, age and their interaction were added as model effects. Next, this model was simplified using stepwise backwards modelling by removing all non-significant (*p* > 0.05) effects. The probability of a higher mortality between the different groups was evaluated by a Cox’s proportional hazard model. Treatment, age, sex and their interactions were added as fixed factors. A post-hoc analysis was performed using risk ratios. Additionally, mortality was visualized using Kaplan–Meier curves.

## 3. Results

### 3.1. Body Weight

There were no significant interactions present. No significant difference in body weight was observed between piglets that were sham drenched and piglets that received no treatment (*p* = 0.203) or between males and females (*p* = 0.441). As expected, the body weight increased over time (day 1: 0.84 ± 0.20 kg, day 3: 0.97 ± 0.28 kg, day 9: 1.76 ± 0.58 kg, day 24: 3.45 ± 1.17 kg, day 38: 5.30 ± 1.74 kg); *p* < 0.001) (Figure 1).

### 3.2. Glucose Levels

There were no significant interactions. There was no effect of age or sex. Glucose levels did not differ between piglets that were sham drenched and piglets that received no treatment (Table 2).

### 3.3. NEFA Levels

There were no significant interactions. Sham drenching had no effect on blood total NEFA levels, nor did the levels differ between male and female pigs. The NEFA levels were significantly lower at the age of 38 days when compared to the age of 9 days (Table 2).

### 3.4. Urea Levels

There were no significant interactions. Urea levels did not differ between piglets that were sham drenched and piglets that received no treatment. Urea levels did not differ significantly between males and females. Urea levels were significantly lower on day 38 compared to day 9 (Table 2).

### 3.5. IgG Levels

There was no significant interaction between treatment and age. Age did not have an effect on IgG levels. IgG levels did not differ between piglets that were sham drenched and piglets that received no treatment (Table 2).

### 3.6. IGF-1 Levels

There was no significant interaction between treatment and age. IGF-1 level was significantly higher on day 38 compared to day 9. IGF-1 levels did not differ between piglets that were sham drenched and piglets that received no treatment (Table 2).

### 3.7. Haematological Analysis

There were no significant interactions. The sex of the animals had no effect on the RBC level, the HCT, the HGB level, the lymphocyte concentration, the monocyte level, the basophil level and the thrombocyte level. However, the total amount of WBCs was significantly higher in males than in females. This increased WBC count in males was due to a higher neutrophil and eosinophil concentration. There was a significant age effect present for the RBCs, the HCT, the HGB, the total WBC count, the lymphocytes, the monocytes and the thrombocytes. There was no age effect on the neutrophil, eosinophil and basophil concentration. No difference was seen between animals that were sham drenched or received no treatment for the RBC level, the HCT, the HGB level, the total WBC count, the lymphocytes, the monocytes, the neutrophils, the eosinophils, the basophils and the thrombocytes (Table 2).

### 3.8. Skin Lesion Scores

There were no significant interactions. No influence of sex was observed (*p* = 0.394). Age did have an effect on the severity of skin injuries (*p* = 0.001). The highest probability of piglets having skin lesions was at the age of 38 days, followed by 24 days, 9 days, 1 day and 3 days. Thus, the post-weaning period posed the highest risk. Piglets that were sham drenched showed no higher risk of having skin lesions compared to piglets that were not treated (*p* = 0.247).

### 3.9. Mortality

The 2-way interactions of treatment*age, sex*age and treatment*sex were not significant (*p* = 0.959, *p* = 0.970 and *p* = 0.696, respectively). There was no effect of sex (*p* = 0.320) or treatment (*p* = 0.619). However, there was a significant effect of age on mortality (*p* < 0.001). All considered piglets had the highest risk of dying on the first day, with this risk decreasing over the following time points. Consequently, mortality was highest during the first 9 days (47.37%), with the most critical moment being the first 3 days (mortality of 34.21%) (Figure 2).

## 4. Discussion

In order for LBW piglets to acquire an adequate amount of energy and nutrients, farmers are often suggested to drench them with milk replacer, colostrum or enriched formula [21,23,24,37,38]. However, supplementing piglets via drenching implies chasing, picking up, restraining and drenching the animals while they are often agitated or scared. It was hypothesized in this study that the act of drenching piglets with a low birth weight, and consequently, a lower energy reserve and chance of survival [39], might aggravate their already weakened situation. If drenching LBW piglets causes an additional burden, it might counteract any potentially positive effect of the supplemented substance. In this respect, the literature data are conflicting, as Declerck and colleagues [39] observed a reduction in mortality when they supplemented LBW piglets (<1.00 kg) with a coconut oil containing booster on the day of birth, whereas an earlier study by Santos et al. [40] did not report an effect on mortality, even though coconut oil was supplemented at a higher total energetic dosage to piglets of a similar birth weight (0.60–0.90 kg) on both the first and second day of life. A later study, in which LBW piglets were supplemented with coconut oil on the day of birth and the second day, also did not show a reduced mortality when comparing supplemented and non-supplemented piglets. It must be mentioned, though, that during the latter study, piglets with a higher birth weight (<1.20 kg) were classified as LBW and the total supplemented energy was lower than the previously mentioned studies [41]. The results of these last two studies support the hypothesis that drenching could nullify a potentially positive effect of the supplemented product. The present study examined the effect of drenching for 7 days after birth vs. non-drenching on the body weight, skin lesion score and survival of exclusively LBW pigs at days 1, 3, 9, 24 (weaning) and 38 (post-weaning).

No difference in body weight was found between LBW piglets that were sham drenched and LBW piglets that were not drenched across the experimental period. Similar results were found in a previous study by de Oliveira et al. [42]. In the latter study, there was no difference in body weight between handled (enforced stroking) and non-handled piglets. Strangely, in this study [42] piglets that were not handled, but exposed to humans, were heavier than their handled littermates. These findings suggest that the performance of piglets, in terms of body weight, is sensitive to the level of exposure or induced stress. This was confirmed in another study showing that the body weight of piglets which experienced human handling as too stressful was affected negatively [43]. What is more, previous exposures to humans that were experienced as negative have been shown to persist in piglets’ memories for up to five weeks. Thus, any interaction with humans could potentially have a detrimental effect on the pig’s performance even beyond weaning [43,44,45]. The results on the body weight of LBW piglets in our study demonstrate that the level of stress caused by drenching does not impact piglet’s growth. Given that the pigs in this study were considered less resilient because of their low body weight and younger age, this lower resilience did not seem to apply to a more profound stressor as drenching. Due to the LBW piglets’ low energy reserves [10,39] and smaller size, they are generally unable to compete against heavier littermates for the better, anterior teats, resulting in an insufficient colostrum intake [2,29,46]. This inadequate consumption of colostrum further depletes the LBW piglets’ energy reserves, which can explain why the animals were not able to struggle much during this experiment’s drenching moments and did not experience this as a significant stressor. Moreover, since LBW piglets are easily outcompeted by their heavier litter mates, it can be assumed that they did not experience previous stressful experiences due to fighting. This can be an additional explanation as to why the LBW piglets did not experience drenching as very stressful, but instead showed rather meek behavior.

In a second part, the present study looked into different biochemical (glucose, NEFA, urea, IgG and IGF-1) and hematological (RBCs, HCT, HGB, WBCs, lymphocytes, monocytes, neutrophils, eosinophils, basophils and thrombocytes) blood parameters after the drenching period (day 9) and two weeks after weaning (day 38) to asses a potential effect of drenching as a chronic stressor. By comparing these blood values between sham drenched and non-drenched LBW piglets, any difference could be attributed to the act of drenching, allowing future supplementation studies to distinguish differences in blood values, due to the act of drenching, from those that are caused by the supplement. Repetitive, unpleasant handling of pigs induces chronic stress and results in elevated corticosteroid, adrenalin and noradrenalin levels [47,48,49]. Corticosteroids, such as the glucocorticoid cortisol, stimulate gluconeogenesis, resulting in higher glucose levels, and inhibit the protein metabolism, resulting in higher protein levels in the blood [50]. Additionally, cortisol has a stimulating role in the urinary excretion of urea [51], and stress increases the circulating levels of NEFAs [52]. If drenching LBW piglets for 7 days were experienced as an unpleasant handling, higher glucose, lower urea and higher NEFA levels would have been expected. However, no differences were seen between the treatment groups. As mentioned before, it should be taken into account that LBW piglets often have low body energy reserves, a reduced ability to mobilize these reserves [10,39,53,54,55] and less energy uptake [56,57]. Thus, it is plausible that the relatively mild stressor (i.e., drenching) was not sufficient enough to mobilize the piglets’ already low glycogen reserves (for a longer period). Moreover, any additional stressor related procedures (i.e., ear tagging, tail docking and castration) had to be considered as well. These interventions have been known to induce pain and cause acute stress, thus increasing the cortisol levels and potentially masking the more subtle effect of drenching-induced stress [58]. However, the blood samplings were performed at least two days after these interventions and cortisol levels are normally only elevated up to 4 h, 2–7 h and 0–1 h after ear tagging, castration and tail docking, respectively [58,59,60,61]. Thus, when considering the effect of these procedures on cortisol and, consequently, glucose, NEFA and urea, no interference was expected at the blood sampling moment at day 9 of life. NEFA and urea levels were both lower at two weeks after weaning compared to immediately after the drenching period. Plausible explanations for these findings could be that neonatal piglets have a higher demand for fatty acid mobilization and gluconeogenesis in response to an increased energy demand, as suggested by Madsen et al. [24], and the high amount of fats that can be found in the sow’s milk [62]. Since this study showed no effect of drenching on the protein or lipid metabolism, nor any increase (or decrease) in glucose levels, and high (or low) blood glucose concentrations are correlated to the chance of survival [49], it can be assumed that drenching did not impose a significant threat on the pigs’ survival chances.

No differences in IgG and IGF-1 levels were found between drenched and non-drenched piglets. These findings suggested that handling LBW piglets did not interfere (negatively) with their suckling behavior. Since colostrum is very rich in immunoglobulins (with IgG being the most abundant) [55] and growth factors [21,63], a reduction in plasma IgG and IGF-1 levels would be expected if the induced stress of drenching had affected the LBW piglets’ suckling. Similar to the biochemical blood analysis, no difference in hematology was seen between sham drenched and non-drenched LBW piglets. These results further confirmed that drenching did not have a negative impact on LBW piglet’s metabolism. There was a higher leucocyte, mainly neutrophil, count in male piglets than in female piglets, regardless of their age or treatment. This increase in males was not due to castration, as this normally results in leukocyte trafficking, during which white blood cells are redistributed to other organs, such as the skin [64]. A plausible explanation would be that the male animals suffered more from infections, which might be represented by the higher, albeit not significant, mortality in male piglets. However, no monitoring of infections was conducted, so no conclusions regarding the reason for the increased leukocyte number could be given.

Skin lesions were scored to determine whether drenching affects the aggression that was experienced by the LBW piglets. Apart from aggressive behavior that is driven by the formation of social or dominance hierarchy, pigs are prone to show more aggression when exposed to stressful circumstances such as low environmental enrichment or mixing with unfamiliar pigs [65]. This was reflected in the present study by the higher presence of skin lesions during the post-weaning period, when unfamiliar piglets were put together in the same pen. It was hypothesized that the additional stress from drenching piglets could potentially increase their aggression as well. On the other hand, de Oliveira et al. [42] suggested that forced human handling could reduce the piglets’ fear. This could increase their ability to cope with stress and reduce intraspecific aggression. In the present study, skin lesions were not affected—positively nor negatively—by drenching. The act of sham drenching might have been more stressful to the piglets than the enforced stroking that was performed in the study by de Oliveira et al. [42], thus it induced more stress and eliminated any positive effect on their behavior. Contrarily, drenching might not have been such a big stressor for the piglets that they responded with more aggression towards their conspecifics. More behavioral and stress research, such as fear and aggression testing (e.g., novel object test, open-field test) and measurements of biomarkers such as salivary cortisol or alpha-amylase, would be required to determine any other effects of drenching on the behavior and stress response of LBW pigs.

In agreement with previous studies, mortality was highest during the first 7 days, with the most critical period being the first 3 days [15,57] for both drenched and non-drenched animals. After this critical first week, mortality declined in the sham drenched group, with only 8% of the animals dying after the drenching period and less than 3% dying after weaning. The non-drenched piglets suffered nearly three times as many losses after the first week (23%), with 15% of the piglets dropping out post-weaning. Given the multifactorial etiology and complexity of mortality in piglets [10], this observation could be important, but requires further research as no statistically significant difference was found in the present study. The sham drenched piglets possibly benefitted from having been picked up and returned to the farrowing crate. During this study, the piglets were often asleep when they were picked up, and thus, awoken and agitated. Putting an empty syringe in their mouth could have activated their suckling reflex and stimulated the piglets to suckle after being returned to the sow. The untreated animals were left alone and remained asleep and therefore missed out on this extra suckling bout, which could explain the difference between these two treatment groups. However, since no difference was seen between drenched and non-drenched animals for the other tested parameters (e.g., body weight), the activation of the piglets by picking them up was probably not enough for them to increase their suckling performance in a matter to gain weight or improve their immunity. During the present study, piglets were not returned in a standardized way (e.g., always at a teat), so confounding was possible. Although there was no significant interaction between sex and treatment, male animals did seem to have lower survival chances when not being handled. In both male and female animals, mortality remained lower than 60% when they were sham drenched. In the untreated group, however, less than 10% of the male piglets survived, while female piglets had survival rates similar to the sham drenched animals. It was already reviewed by Muns et al. [10] that male piglets were more likely to die before weaning even if they were heavier. Even though male piglets are often born heavier than females, a male-biased susceptibility to mortality seems to exist. The higher energy demands in male piglets could be attributed to secondary sexual characteristics, such as larger body size and tusks, resulting in poorer thermoregulatory abilities and decreased immunocompetency [66]. The results of the present study suggested that male LBW animals could benefit from being handled during the neonatal period, while female animals might not experience any positive effect from early human handling. Further research to determine this potentially sex-biased effect is required before applying different management strategies for LBW piglets, depending on their sex.

## 5. Conclusions

Sham drenching did not affect LBW piglets’ performance or mortality during the drenching period, the suckling period and after weaning. Thus, drenching can be applied safely in underprivileged piglets as an intervention to enhance their survival chances. For studies examining the effects of supplements, it is advised to always incorporate a non-drenched group into the experimental set-up. Consequently, it will be possible to attribute any observed effect to either the supplemented product or the act of drenching.

## Figures and Tables

**Figure 1 animals-11-00404-f001:**
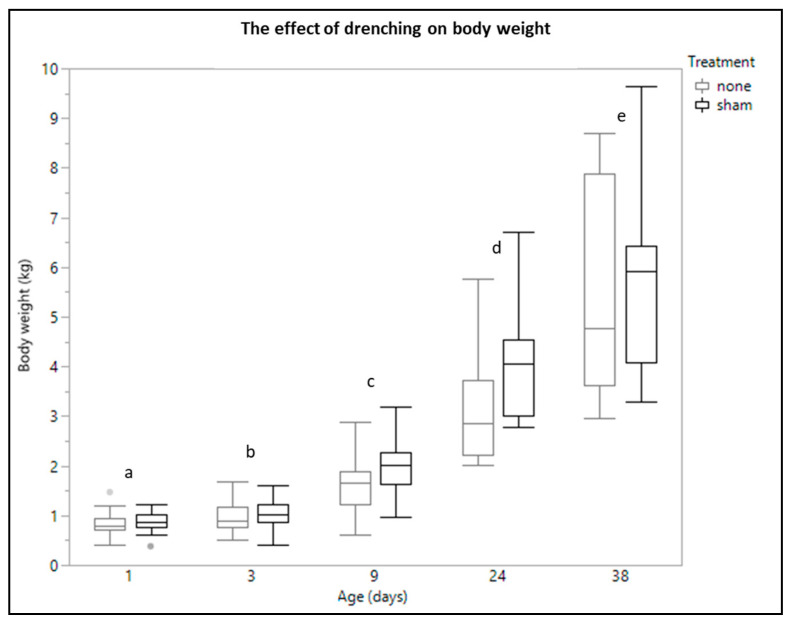
Boxplot of body weight (*n* = 76) from low birth weight piglets that received no treatment (*n* = 39) or were sham drenched (*n* = 37) at different time points (day 1 (*n* = 76), day 3 (*n* = 50), day 9 (*n* = 40), day 24 (*n* = 35) and day 38 (*n* = 28)). Significant age differences (linear mixed models, Tukey post-hoc analysis, *p ≤* 0.05) are indicated by a different letter.

**Figure 2 animals-11-00404-f002:**
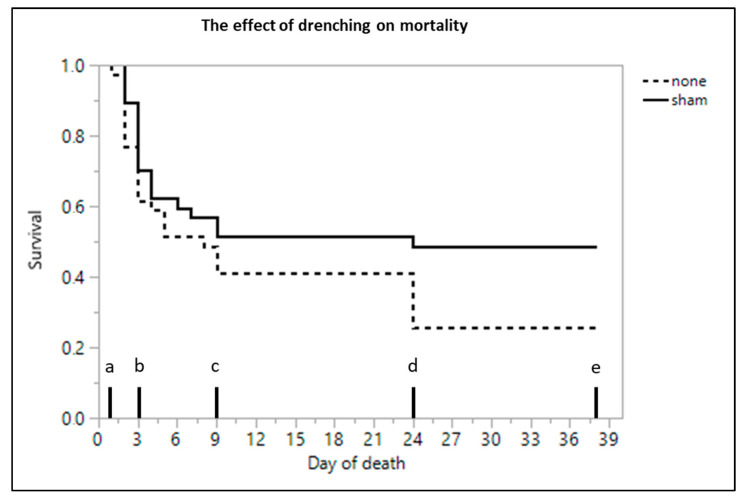
Cumulative mortality of low birth weight piglets that were sham drenched (*n* = 37) or received no treatment (*n* = 39) over time. Significant differences (Cox’s proportional hazard model, Kaplan–Meier survival plot, *p ≤* 0.05) between time points are indicated by a different letter.

**Table 1 animals-11-00404-t001:** Nutrient and chemical composition of sows’ gestation and lactation diets.

Composition	Gestation Diet	Lactation Diet
Crude protein (%)	13.9	15.0
Crude fat (%)	3.8	3.8
Crude fiber (%)	8.3	7.4
Crude ash (%)	5.1	5.4
Total sugars and starch (%)	40.0	41.2
Lysine (%)	0.7	0.8
Methionine (%)	0.3	0.3
Phosphor (%)	0.5	0.6
Calcium (%)	0.7	0.8
Vitamin E (mg/kg)	150.0	150.0
Vitamin A (IU/kg)	10,000	10,000
Vitamin D3 (IU/kg)	2,000	2,000
Iron (mg/kg)	53.0	53.0
Iodine (mg/kg)	2.0	2.0
Copper (mg/kg)	5.0	5.0
Manganese (mg/kg)	43.0	43.0
Zinc (mg/kg)	15.0	15.0
Selenium (mg/kg)	0.4	0.4

**Table 2 animals-11-00404-t002:** Blood values (median ± SD) of glucose, non-esterified fatty acids (NEFA), urea, immunoglobulin G (IgG), insulin-like growth factor 1 (IGF-1), red blood cells (RBCs), hematocrit (HCT), hemoglobulin (HGB), white blood cells (WBCs), lymphocytes, monocytes, neutrophils, eosinophils, basophils and thrombocytes, presented by age, sex and treatment from selected low birth weight piglets (linear mixed models, Tukey post-hoc analysis, *p ≤* 0.05).

Dependent Variable	Age					Sex					Treatment				
	Day 9		Day 38			Female		Male			No treatment		Sham		
	*n*	Median ± SD	*n*	Median ± SD	*p*-Value	*n*	Median ± SD	*n*	Median ± SD	*p*-value	*n*	Median ± SD	*n*	Median ± SD	*p*-Value
**Glucose (mmol/L)**	20	6.87 ± 1.64	16	6.40 ± 1.70	0.300	24	6.40 ± 1.92	12	6.75 ± 1.03	0.960	16	6.55 ± 2.02	20	6.56 ± 1.32	0.341
**NEFA (mmol/L)**	20	0.58 ± 0.79	14	0.17 ± 0.18	<0.001	23	0.46 ± 0.47	11	0.45 ± 1.05	0.925	15	0.47 ± 0.52	19	0.33 ± 0.82	0.125
**Urea (mmol/L)**	19	3.66 ± 0.88	15	1.66 ± 0.69	<0.001	22	2.84 ± 1.41	12	2.32 ± 1.13	0.479	14	3.24 ± 1.13	20	2.32 ± 1.40	0.081
**Ig G (mg/mL)**	11	3.31 ± 4.93	11	2.97 ± 1.02	0.344	16	3.24 ± 4.27	6	2.55 ± 1.13	-	10	3.39 ± 5.50	12	2.55 ± 1.79	0.560
**IGF-1 (ng/mL)**	11	8.95 ± 15.99	11	38.61 ± 21.02	0.008	16	23.48 ± 25.05	6	15.03 ± 14.48	-	10	23.48 ± 27.05	12	20.17 ± 17.62	0.175
**RBC (1012/L)**	10	4.27 ± 0.42	15	5.69 ± 0.56	<0.001	17	5.23 ± 0.96	8	5.33 ± 0.89	0.687	9	5.11 ± 1.04	16	5.46 ± 0.88	0.769
**HCT (%)**	10	30.45 ± 2.50	15	35.9 ± 3.51	0.005	17	33.5 ± 3.58	8	32.30 ± 4.96	0.600	9	32.60 ± 3.88	16	33.20 ± 4.15	0.759
**HGB (g/dL)**	10	8.40 ± 0.71	15	10.40 ± 1.12	0.001	17	9.20 ± 1.43	8	9.35 ± 1.26	0.814	9	9.20 ± 1.70	16	9.65 ± 1.17	0.418
**WBC (103/µL)**	10	10.73 ± 5.10	15	17.95 ± 4.20	0.021	17	13.18 ± 5.61	8	18.24 ± 4.11	0.039	9	12.66 ± 6.58	16	17.81 ± 4.39	0.908
**Lymphocytes (103/µL)**	10	4.18 ± 1.75	15	7.96 ± 1.62	0.001	17	6.15 ± 2.42	8	7.06 ± 1.87	0.616	9	4.94 ± 2.82	16	6.93 ± 1.90	0.950
**Monocytes (103/µL)**	10	0.89 ± 0.37	15	1.55 ± 0.57	<0.001	17	0.94 ± 0.52	8	1.47 ± 0.70	0.086	9	1.08 ± 0.51	16	1.12 ± 0.64	0.952
**Neutrophils (103/µL)**	10	5.83 ± 3.15	15	8.33 ± 2.90	0.172	17	6.14 ± 3.22	8	9.70 ± 2.24	0.005	9	5.49 ± 3.77	16	8.40 ± 2.52	0.612
**Eosinophils (103/µL)**	10	0.10 ± 0.20	15	0.19 ± 0.13	0.817	17	0.13 ± 0.11	8	0.27 ± 0.20	0.038	9	0.12 ± 0.12	16	0.19 ± 0.17	0.913
**Basophils (103/µL)**	10	0.02 ± 0.06	15	0.01 ± 0.02	0.177	17	0.01 ± 0.05	8	0.02 ± 0.01	0.552	9	0.01 ± 0.06	16	0.02 ± 0.01	0.440
**Thrombocytes (103/µL)**	10	1087.50 ± 374.48	15	444.00 ± 173.61	<0.001	17	646.00 ± 458.02	8	560.00 ± 283.45	0.461	9	764.00 ± 535.69	16	596.00 ± 334.41	0.689

## Data Availability

All data presented in this study is contained within the article.

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
