# Peer review of "Handling Associated with Drenching Does Not Impact Survival and General Health of Low Birth Weight Piglets"

_animals, 2021, doi:10.3390/ani11020404_

Round 1

Reviewer 1 Report

This manuscript used piglets from 67 sows to evaluate effects of drenching on their health, and found the drenching does not impact survival and general health of low birth weight piglets. It is an interesting topic. These results are important for the practice. The manuscript could be accepted in this revised version.

Author Response

The authors would first of all like to thank the reviewers for the constructive comments. We have made every effort to include and address all concerns that were pointed out during the previous (manuscript 900790) and current resubmission (manuscript 1033398). We hope the improvements to the manuscript are to your liking, but if further clarifications or actions on our part are required, we are at your disposal.

Reviewer 2 Report

This manuscript looks at the stress the act of drenching causes low birth weight piglets to determine if procedure stress would offset the clinical aspect of supplementing piglets. 

Ln 96 what does skin lesions have to do with a drenching stress? Justify/explain reason for collecting data

Ln 103 Why did you not use a control solution? Half the struggle during drenching is actually getting the liquid to the animal. That would mimic gut fill, the fact that they fight when administering liquid not only when pick them up, and the potential for any aspiration. All huge factors in contributing to drenching stress.

Ln 120 why did you no take a baseline blood sample? Explain in this section the reason behind the parameters. Why did you not measure cortisol if this is a stress study?

Ln 138 why did you report everything in SD not SE?

Results there are times you write out the values in the text when it's already presented in tables (e.g. haematology numbers) and other times you don't give any values anywhere (e.g. skin lesions). All values should be in the manuscript at least once but be sure they don't repeat. Check this throughout.

Figures define acronyms so that figures can stand alone

Discussion

Ln 320 what type of infections are males more predisposed to just for gender?

Ln 323 the scenarios you discuss here are a lot different stressors than a quick clinical procedure stress. I can't imagine drenching stress would make pigs more aggressive (ln 327) so that and doing skin lesions does seem a far-fetched measure. You also showed no differences in skin lesions don't think you can make this assumption here.

Ln 343 I agree with this statement however, you found no treatment differences so be sure to indicate that here. This statement goes for the conclusions as well (Ln377) that's a strong conclusion when no differences were found in mortality.

Author Response

This manuscript looks at the stress the act of drenching causes low birth weight piglets to determine if procedure stress would offset the clinical aspect of supplementing piglets. 

Ln 96 what does skin lesions have to do with a drenching stress? Justify/explain reason for collecting data

The authors are grateful to the reviewer to bring this to their attention, as we acknowledge the direct link between skin lesions and drenching-related stress is not very obvious. The skin lesion scoring was incorporated into this study, because it was also applied during another (unpublished) study that focused on the effect of different, supplemented bioactive compounds. In the latter study, skin lesions scores were incorporated, following the hypothesis that the supplement might provide the LBW piglet with additional energy, allowing it to demonstrate more competitive, aggressive behavior. To ensure that any difference in skin lesions would be attributable to the supplement and not to the drenching act, skin lesion scoring was applied in the current study. Based on the results of this study (no effect of handling on skin lesions), readers can safely assume that any difference in skin lesions, after supplementation, is due to the supplement.

We have explained this more in the manuscript to clarify the motives behind skin lesion score for the reader:

Lines 122-128: Skin lesion scoring has been validated as an indicator for aggressive behavior [1]. It is often applied in studies that focus on supplementation or diets [2,3], following the hypothesis that the supplement might provide the piglet with additional energy, allowing it to demonstrate more competitive, aggressive behavior. To determine whether handling (drenching) LBW piglets had an effect on their competitive behavior, and consequently, to ensure that any observed difference in skin lesions during future supplementation studies can be attributed to the supplement and not the act of drenching, each LBW piglet was scored for skin lesions on the snout and skin.

Ln 103 Why did you not use a control solution? Half the struggle during drenching is actually getting the liquid to the animal. That would mimic gut fill, the fact that they fight when administering liquid not only when pick them up, and the potential for any aspiration. All huge factors in contributing to drenching stress.

We would like to refer for this matter to the revised version of the manuscript (which you have reviewed) and to the rebuttal letter sent to the reviewers of our first submission. We recapitulate our response to this question which was raised earlier below:

The purpose of this study was to tackle the knowledge gap concerning the effect of drenching weaker, low birth weight piglets. In most studies, liquids are supplemented without considering a potential effect of the handling that is associated with drenching, often resulting in different, inconsistent results. Therefore, we wanted to study whether the act of drenching might influence the health and survival of low birth weight piglets, and consequently, the effect of supplemented liquids. We completely agree that the use of a control group in which a liquid (often the solvent of the studied compound, such as water, milk…) is supplemented, is the best way to evaluate the effect of a supplemented substance. However, for the evaluation of the drenching act, we have deliberately opted for a sham group, and no third control group with liquid drenching, in our experimental design for the two following reasons:

  • Drenching includes catching, fixating and supplementing a liquid substance into the mouth, rather than a forced administration into the throat. Consequently, the animal should not experience any suffocation due to the administered liquid which should have minor to none influence on the experienced stress, in contrast with e.g. gavage. Therefore, the authors wanted to focus on the most important stressor of drenching, the handling, which could be replicated by a sham drenching with an empty syringe.
  • The authors wanted to avoid any influence of a liquid to be able to evaluate the effect of the drenching act. By adding a third control group in which the animals would be supplemented with a liquid, only the effect of the liquid would be observed when comparing this group to the sham group. Supplementing a liquid as a control could influence the animal by, e.g. filling up the stomach and creating a false feeling of saturation. Adding a liquid as a control would only be useful when it would function as a control for another supplemented compound and not the act of drenching.

Ln 120 why did you no take a baseline blood sample? Explain in this section the reason behind the parameters. Why did you not measure cortisol if this is a stress study?

We have not taken a baseline blood sample to avoid having to sample more blood from the already weakened low birth weight piglets of from more piglets than necessary in the current set-up for ethical reasons. The ‘no treatment’ group consisted of piglets that were not sham drenched and acted as a control group for the blood values when the blood values of sham drenched piglets were analyzed.

The reason behind the parameters has been clarified more in the text:

Lines 290-297: In a second part, the present study looked into different biochemical (glucose, NEFA, urea, IgG and IGF-1) and hematological (RBC, HCT, HGB, WBC, lymphocytes, monocytes, neutrophils, eosinophils, basophils and thrombocytes) blood parameters after the drenching period (day 9) and two weeks after weaning (day 38) to asses a potential effect of drenching as a chronic stressor. By comparing these blood values between sham drenched and non-drenched LBW piglets, any difference could be attributed to the act of drenching, allowing future supplementation studies to distinguish differences in blood values, due to the act of drenching, from those that are caused by the supplement.

Regarding the determination of cortisol in blood samples, we would like to point out that this study is not set-up as a stress study. The intention was to document the effects of the act of drenching on the health and performance of underprivileged piglets. Nevertheless, we agree that a biomarker for the stress induced by certain manipulations could shed a light on the mode of action of effects on health or performance. However, we dispute that cortisol levels in blood suit this purpose as cortisol in blood is not considered to be a good marker. Cortisol release into the blood occurs very quickly and is, consequently,  high due to the blood sampling itself. NEFA and glucose are associated with the cortisol response but respond later and since we already had the blood for other parameters we could have a look to see whether this could have an indication about the stress response without having false results. Thus, we opted to measure other blood parameters such as NEFA and glucose which we – although indirectly – could relate to a cortisol response. We have considered using saliva and hair samples for cortisol measurements. This would require a saliva sample of at least 200 µL, preferably daily, before and after sham drenching from the day of birth until day 7. This would be practically impossible, since saliva production is very low in neonatal piglets, and even lower in low birth weight piglets. Moreover, the collection of saliva samples from these young animals would not be possible without fixating the animal and therefore causing stress. Hair samples could be used to measure chronic stress over a longer period. Short, repetitive stress – induced by drenching during 7 days – would not be visible in hair samples. Moreover, the newborn piglets would have had to be shaved to create a cortisol baseline of the hair and shaved again one week later which would result in an insufficient amount of collected hair and the possible effect of drenching still at the epidermal, non-shavable level [4]. As mentioned in the manuscript, further research, such as open-field testing or novel object testing, is required to obtain more information about the effect of sham drenching on behavior and stress.

We have added the mentioning of cortisol to this section to inform the reader that this was not a stress focused study and more research is required:

Lines 352-355: More behavioral and stress research, such as fear and aggression testing (e.g. novel object test, open field test) and measurements of biomarkers such as salivary cortisol or alpha-amylase would be required to determine any other effects of drenching on the behavior and stress response of LBW pigs.

Ln 138 why did you report everything in SD not SE?

Standard deviation was chosen to illustrate the amount of variability within our sample, because there was a relatively large sample size (76 animals in total, > 35 animals per treatment). Therefore, standard deviation was chosen over standard error, as the latter would take the large sample size into account, and subsequently, be much smaller. This could wrongly be interpreted by the reader as a small variation. Thus, we chose SD as a better descriptive of our sample distribution than SE.

Results there are times you write out the values in the text when it's already presented in tables (e.g. haematology numbers) and other times you don't give any values anywhere (e.g. skin lesions). All values should be in the manuscript at least once but be sure they don't repeat. Check this throughout.

Figures define acronyms so that figures can stand alone

All abbreviations have now been explained in the figures and tables, so they can stand alone.

Discussion

Ln 320 what type of infections are males more predisposed to just for gender?

It was not the authors’ intention to attribute the higher leukocyte number in males to sex-related infections. The leukocytes might have been increased in males by infections that were not sex-related. The occurrence, and consequently, the etiology of infections was not monitored, so it was impossible to determine whether there were more infections in males and whether these infections were sex-dependent, a result from the castration…

We have added the following to this section to ensure that the reader can interpret this correctly:

Lines 334-337: A plausible explanation would be that the male animals suffered more from infections which might be represented by the higher, albeit not significant, mortality in male piglets. However, no monitoring of infections was conducted, so no conclusions regarding the reason of the increased leukocyte number could be given.

Ln 323 the scenarios you discuss here are a lot different stressors than a quick clinical procedure stress. I can't imagine drenching stress would make pigs more aggressive (ln 327) so that and doing skin lesions does seem a far-fetched measure. You also showed no differences in skin lesions don't think you can make this assumption here.

Skin lesion scoring was incorporated into this study to ensure that handling, associated with drenching did not influence the animals’ behavior to ensure a correct attribution of effect in future studies that focus on supplementation. The latter studies could use skin lesion scoring to observe any changes in aggressive or competitive behavior, since supplementation of, e.g. an energy booster, might make the piglets more resilient, and thus, more competitive. No direct effect of drenching associated stress (which would probably be a meeker than a more aggressive response) was expected by the authors.

Ln 343 I agree with this statement however, you found no treatment differences so be sure to indicate that here. This statement goes for the conclusions as well (Ln377) that's a strong conclusion when no differences were found in mortality.

The authors agree with the reviewer that the text should indicate clearly that there was no significant difference to avoid any wrong interpretations by the reader. Therefore, the following clarification was added:

Lines 359-363: The non-drenched piglets suffered nearly three times as much losses after the first week (23%), with 15% of the piglets dropping out post-weaning. Given the multifactorial etiology and complexity of mortality in piglets [5], this observation could be important, but requires further research as no statistically significant difference was found in the present study.

The conclusion was also rephrased (cfr. suggestion by reviewer 3):

Lines 388-393: Sham drenching did not affect LBW piglets’ performance or mortality during the drenching period, the suckling period and after weaning. Thus, drenching can be applied safely in underprivileged piglets as an intervention to enhance their survival chances. For studies examining the effects of supplements, it is advised to always incorporate a non-drenched group into the experimental set-up. Consequently, it will be possible to attribute any observed effect to either the supplemented product or the act of drenching.

Reviewer 3 Report

The Manuscript entitled “Handling associated with drenching does not impact survival and general health of low birth weight piglets” is interesting and fits the scope of the Journal. In my opinion the Title of the Manuscript should be corrected because does not encourage to the perusal of the text. The Title contain conclusion.

Remaining Sections are well written, only Conclusions require rephrasing, because it is rather the Summary not Conclusions.

Author Response

The Manuscript entitled “Handling associated with drenching does not impact survival and general health of low birth weight piglets” is interesting and fits the scope of the Journal. In my opinion the Title of the Manuscript should be corrected because does not encourage to the perusal of the text. The Title contain conclusion.

We agree that the title expresses the main conclusion of the manuscript. However, we hope that this encourages the reader to figure how we came to that conclusion and which parameters were studied. We could turn the title into a question (Does the handing associated with drenching impact survival and general health in low birth weight piglets?). However, we would like to refer for this matter to our previous response to the reviewers who assessed our 1st version. They pointed out that the original title ‘Drenching as a tool to improve survival in low birth weight piglets:
evaluating the effects of the procedure’ created wrong expectations for readers in a sense as this would refer to supplementation with liquids, rather than handling piglets. We felt that the title of this submission clearly pointed out what we did and what is the main conclusion to not create false expectations. We have also opted for this title structure, following the guidelines of the journal and the example of other articles, recently published in Animals, that have also chosen a conclusive title
[6-10].

Remaining Sections are well written, only Conclusions require rephrasing, because it is rather the Summary not Conclusions.

The authors agree with the reviewer and have altered the discussion:

Lines 388-393: Sham drenching did not affect LBW piglets’ performance or mortality during the drenching period, the suckling period and after weaning. Thus, drenching can be applied safely in underprivileged piglets as an intervention to enhance their survival chances. For studies examining the effects of supplements, it is advised to always incorporate a non-drenched group into the experimental set-up. Consequently, it will be possible to attribute any observed effect to either the supplemented product or the act of drenching.

Reviewer 4 Report

The authors have conducted a study investigating sham treatment on no treatment. I have to admit I fail to understand from the introduction why this putting a syringe into a piglets mouth would be a problem and cause extra stress compared to handling them for castration, teeth clipping, injections etc that you describe in line 86:

Line 86: before the age of  one week, all piglets were ear tagged, tail docked, received a 200 mg iron dextran injection and all male piglets were castrated using meloxicam analgesics.

And also the blood sampling would surely be more stressful than sham feeding?

Surely to test this the piglets should not have had any other stressors applied to them? Has any previous literature suggested that putting a syringe into a piglets mouth is detrimental? Maybe if it was tube feeding there could be a difference? As the tube could penetrate the esophagus. I am sorry I think I just really have either missed the point or need the introduction more clarified. All though you have made a lot of analyses I think it would be better published as a technical note as there are so few differences (and you don’t know if they are in fact sham handling related – as all piglets have been handled for normal procedures)?

Minor details:

Line 16 – suggst in modern pig production

Line 36: suggest MLBW

Line 39 – is this enough?

Line 202 -217 – just refer to the table – no need to present all the data again.

Author Response

The authors have conducted a study investigating sham treatment on no treatment. I have to admit I fail to understand from the introduction why this putting a syringe into a piglets mouth would be a problem and cause extra stress compared to handling them for castration, teeth clipping, injections etc that you describe in line 86:

Line 86: before the age of one week, all piglets were ear tagged, tail docked, received a 200 mg iron dextran injection and all male piglets were castrated using meloxicam analgesics.

And also the blood sampling would surely be more stressful than sham feeding?

Surely to test this the piglets should not have had any other stressors applied to them? Has any previous literature suggested that putting a syringe into a piglets mouth is detrimental? Maybe if it was tube feeding there could be a difference? As the tube could penetrate the esophagus. I am sorry I think I just really have either missed the point or need the introduction more clarified. All though you have made a lot of analyses I think it would be better published as a technical note as there are so few differences (and you don’t know if they are in fact sham handling related – as all piglets have been handled for normal procedures)?

We agree that procedures, such as castration, ear tagging or injections, are significant stressors that will most likely induce more acute stress in piglets than the handling that is associated with sham drenching. As brought to our attention by the reviewer, the introduction could clarify this more. The authors have added information in the hypothesis to ensure more clarification for the reader. We would also like to explain to the reviewer why, during this study, these procedures were deliberately kept into the experimental design:

  • The experiment was set up as a field research, mimicking normal farming circumstances and providing an in-dept research on the possible effect of handling very young piglets during drenching, in addition to normal farming management.
  • Observing the potential effect, associated with drenching, under more controlled circumstances (e.g. no castration) would indeed allow us to observe effects that are solemnly attributable to handling piglets during drenching, but the goal of this experiment was to question the efficiency and differences that are often found in literature (in which normal piglet husbandry procedures are often still performed, regardless of the involved supplementation or handling [11-15]) and farming circumstances by observing the intervention that is often applied, i.e. drenching, under normal circumstances. By restricting the experimental design to only sham drenching, any observed effect (positive or negative) could potentially be nullified or increased when – based on the found results – it would be implemented afterwards in study designs that contain normal farming procedures or at farms that aim for a good farming management. The experimental design was, thus, deliberately set-up to create a model in which the handling, associated with drenching, was the only deviating factor from normal farming management and the straw that broke the camel's back of these already small and weakened piglets.
  • The period, prior to the husbandry procedures, was included and analyzed in the current study. By drenching the piglets from birth up to 6 days of age, any potential effect of handling piglets during drenching could be observed during the first week (before castration and teeth clipping), from the second week until weaning and after weaning until the age of 38 days. However, no effect of handling very young, low birth weight piglets that, contrarily to normal weight piglets, might not be able to cope with both painful husbandry procedures and drenching, was observed during the first week (or later). Similarly, there was no difference in body weight or mortality between males and females which suggested that castration was not the main reason for the absence of any effect of sham drenching.
  • The effect of normal husbandry procedures, such as castration and tail docking, are known to induce an acute stress response of which the effects – based on literature – were not expected to influence the more repetitive (7 days) stressor, sham drenching, as mentioned in lines 309-315.

As mentioned and suggested by the reviewer, we have clarified the introduction with additional information regarding the experimental design and hypothesis:

Lines 71-80: Drenching usually involves catching and fixating the animals, thus, potentially acts as a stressor. Given the LBW piglets’ vulnerable condition, this potential stressor might further aggravate their chance of survival and negatively influence their health and survival, rather than provide the animals with additional energy or bioactive compounds. Consequently, it could be expected that sham drenching LBW piglets daily during their first week of life, might result in higher mortality rates when these weakened animals are confronted with normal husbandry procedures that act as acute stressors, such as castration or teeth clipping. Therefore, it was hypothesized in this study that the act of drenching would provoke additional stress and affect LBW piglets negatively, resulting in a worsened performance (body weight, general (blood) health parameters) and a higher pre-weaning mortality.

The authors agree that this manuscript contains a substantial amount of technical content and will most likely appeal to readers that can use this paper’s results to implement them in supplementation studies or farming management. As mentioned by the reviewer, the authors have conducted many analyses, to ensure an in-depth research about handling, associated with drenching. Although, the authors agree that a technical note could also apply for this manuscript, the limitation of 5,000 words would imply removing a lot of information or some of the analyses. This, given the many analyses and complexity of the subject, might not enable the reader to receive enough information concerning the experiment’s design, conducted tests or results. Moreover, publishing all results from the current study may prevent unnecessary replication of the tests/experiments performed here. Therefore, we would still favor a publication as article.

Minor details:

Line 16 – suggest in modern pig production

The suggestion has been implemented: the word ‘porcine’ has been replaced with ‘pig’.

Line 36: suggest MLBW

We agree that this abbreviation simplifies the text for the reader and has been implemented in both the abstract and material & methods.

Lines 36-37: The mean litter birth weight (MLBW) and standard deviation (SD) were calculated. LBW piglets (n = 76) were defined as weighing between (MLBW-1*SD) and (MLBW-2.5*SD).

Lines 103-105: For each litter, the mean birth weight (MLBW) and SD were calculated. Piglets with a birth weight of MLBW-2.5*SD < piglet birth weight < MLBW-1*SD were characterized as ‘LBW piglets’.

Line 39 – is this enough?

Based on preliminary testing with volumes of 2 mL of milk, the average catching and drenching time was 29.6 sec ± 8.1 SD per piglet (average catching time by 1 person: 10.5 sec ± 5.9; average drenching time: 19.0 sec ± 5.7). Therefore, every piglet was sham drenched during 20 seconds.

This information was withheld from the abstract due to the limitation on words, but explained in the material & methods section (lines 115-118).

Line 202 -217 – just refer to the table – no need to present all the data again.

All data, that can be found in the table, has been removed from the text.

Round 2

Reviewer 4 Report

Dear authors, thank you for clarifying your thoughts on the design. While I still do not completely agree it is more clear now in the introduction why an on-farm study on this topic is relevant. 

I now only have minor: p-values have a lot of ciffers - 3 is enough? P<0.001 or P=0.956

I can see you have changed some of the authors in the discussion to listing all the author names - does animals not just normally want one author name and et al?  

Author Response

We would like to thank the reviewer for the previous assessment and useful input.

We agree that is more common to only us 3, instead of 4, decimals when mentioning P-values. Therefore all P-values have been adjusted to 3 decimal digits.

All references to papers in the text have been altered as well, following the reviewer’s suggestion: only 1 author name, followed by ‘et al.’ when there is more than 1 author, is used in the text.

This manuscript is a resubmission of an earlier submission. The following is a list of the peer review reports and author responses from that submission.

Round 1

Reviewer 1 Report

Tichelen et al report here a well-conducted study that drenching as a tool to improve survival in low birth weight piglets: evaluating the effects of the procedure. It is an interesting topic. This reviewer, however, has a few suggestions that would improve the manuscript before population.

  1. Line 26 and 36, What “sham”mean? The author need to describe it clear when it first appeared. 
  2. The experiment design need to be more specific in the abstract. 
  3. Please add the table note for table 1 and 2. 
  4. p should be italic in the figure legends. 

Reviewer 2 Report

This manuscript is very well-written and appears to be justified in its scientific nature to determine if the "drenching procedure" had an effect on low birthweight piglets.  The study was designed and conducted well - the reviewer feels that it would have been interesting to use a liquid or substrate for the "drenching procedure" as the liquid introduction and being swallowed (and hopefully not aspirated) may attribute to stress on the piglet.

Nonetheless, the current study describes the objectives well and presented the data and subsequent conclusions that support the results.  No additional corrections are needed.

-What are the main claims of the paper and how significant are they?

The main claims of the paper are that the physical act of drenching  does not cause any undue stress on low birthweight piglets.  It is  rather significant to test this theory in that handling a pig (and

especially a low birthweight piglet) and attempting to drench it could cause more stress than the benefit of drenching.  This study does test this theory. –

How does the paper stand out from others in its field?  

This paper stands out from others in the field in that it is testing a "sham procedure" against a non-stressful event to see if there are any  etrimental effects.  Once we know that this is not harmful to these piglets, we can concentrate more on what supplements we can provide to them for their improved survival. –

Are the claims novel? If not, which published papers compromise novelty?

Yes, the papers claims are novel.

Are the claims convincing? If not, what further evidence is needed? 

The claims are rather convincing - it should be noted that it was a "sham procedure" instead of the physical act of drenching with liquid which at times can provide a novel stressor to the piglet - maybe another study with no handling, sham procedure and one with liquid.

Are there other experiments or work that would strengthen the paper further?  Listed above

How much would further work improve it, and how difficult would this be? Would it take a long time? Listed above -> it may take a fair bit of time as this study was conducted with over 60 farrowings.

Are the claims appropriately discussed in the context of previous literature?

 Yes, the literature supports the claims.

Reviewer 3 Report

First I have a question: what is drenching for you?

If I good understand what you were writing about it was only "putting an empty syringe in piglets' mouth" - so for me it is only manipulation, drenching require administration of some liquid (could be only water, but more often something more). So the title is not adequate to the content. To evaluate the procedure in the presented experiment we need the third group where LBW piglets will be really drenched.

Presented results (I think about the title) could be the preliminary study, it is not enough for the complete manuscript.